# Decouple and Reconstruct: Mining Discriminative Features for Cross-domain Object Detection

## Abstract

In recent years, a great progress has been witnessed for cross-domain object detection. Most state-of-the-art methods strive to handle the relation between local regions by calibrating cross-channel and spatial information to enable better alignment. They succeed in improving the generalization of the model, but implicitly drive networks to pay more attention on the shared attributes and ignore the domain-specific feature, which limits the performance of the algorithm. In order to search for the equilibrium between transferability and discriminability, we propose a novel adaptation framework for cross-domain object detection. Specifically, we adopt a style-aware feature fusion method and design two plug-and-play feature component regularization modules, which repositions the focus of the model on domain-specific features by restructuring the style and content of features. Our key insight is that while it is difficult to extract discriminative features in target domain, it is feasible to assign the underlying details to the model via feature style transfer. Without bells and whistles, our method significantly boosts the performance of existing Domain Adaptive Faster R-CNN detectors, and achieves state-of-the-art results on several benchmark datasets for cross-domain object detection.

## 1 Introduction

Along with the advances in deep convolutional neural networks (Krizhevsky et al., 2012), remarkable progresses have been achieved in various visual tasks (He et al., 2016; Long et al., 2015; Ren et al., 2017). Such outstanding achievements heavily rely on the assumption that training and test data are drawn from an independent identical distribution. Nevertheless, due to the changes of environmental condition in real-world, a phenomenon known as "domain shift" occurs and challenges this assumption, thus leading to a significant performance decay. Although collecting more training data can alleviate such problems to some extent, it is non-trivial because of the high cost of image annotation.

An appealing solution to alleviate the shift is introducing domain adaptation (DA) methodology, which aims at bridging the gap between domains by learning invariant representations. In this paper, we focus on the unsupervised scenarios, called unsupervised domain adaptation (UDA), where the training set is formed from labeled source data and unlabeled target data.

Recently, a variety of UDA methods have been well-studied for classification (Tzeng et al., 2014; Ganin & Lempitsky, 2015; Saito et al., 2018; Wang et al., 2020; Li et al., 2021b) and semantic segmentation (Hoffman et al., 2018; Zou et al., 2018; Dong et al., 2020; Li et al., 2020; Jaritz et al., 2021), which are mostly based on minimizing domain discrepancy measurement or adversarial training. These strategies have made great progress but are hard to be applied to object detection directly because of the local nature of detection tasks (Zhu et al., 2019). To address the adaptation dilemma for detection tasks, instance-level feature alignment has been widely used in Chen et al. (2018b); Saito et al. (2019); Zheng et al. (2020). Most of them strive to handle the relation between local regions by calibrating cross-channel and spatial information. However, it may implicitly drive networks to pay more attention on shared attributes and ignore the domain-specific feature. As shown in Figure 1, DA reduces the domain shift by feature alignment between source and target

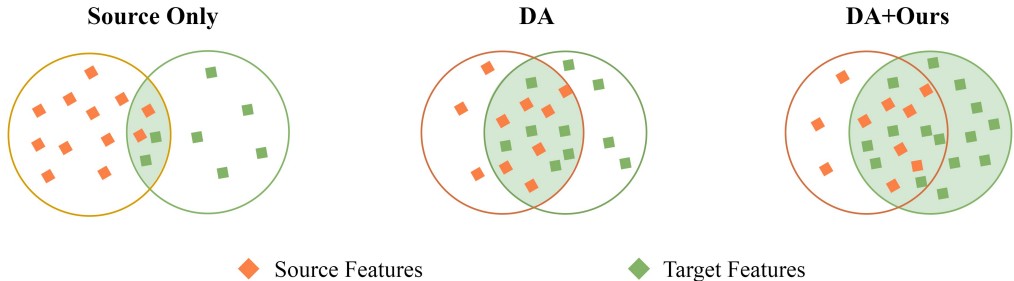

Figure 1: Illustration of the evolution process of source and target feature distributions. Source Only: feature distributions obtained by the model without domain adaptation. DA: aligned feature distributions after domain adaptation. DA+Ours: refined feature distributions by applying our approach to the DA model. Compared to Source Only, DA reduces the domain gap and explores the domain-invariant features by adversarial training at the cost of discriminative features in source and target domain. Our supplementary framework highlights the target-like discriminative feature, which leading to better adaptive performance.

domain, which highlights the common components at the cost of domain-specific features. We review the feature alignment of domain adaptation and suppose that image-level style and instance-level content play a vital role in detection tasks. The style information should be globally consistent within its own domain while local homogeneous content information between different domains ought to be analogous, and vice versa.

Building upon the above findings, we propose a novel adaptation framework for cross-domain object detection, which supplements the domain-specific representation by restructuring the content and style information. Specifically, we adopt the *Domain Adaptive Instance Normalization* (DAdaIN) on shallow layers to generate synthesis features, which contributes on encoding the domain-specific and domain-invariant features in source and target domain. Furthermore, *Global Style Alignment* (GSA) module and *Local Content Alignment* (LCA) module are exploited to regularize the consistency of style and content between multiple domains. GSA consists of two sub-modules, i.e., style similarity alignment and style consistency alignment, which are adopted in multi-layer to align global style representation. Moreover, LCA is proposed to calibrate the content information of local regions, which enhances the model's response to foreground objects of various styles.

In summary, the main contributions of this work are three-fold:

- We review the feature alignment in domain adaptation and point out that it may implicitly drives networks to focus more on public component and ignore the domain-specific feature. Furthermore, we suppose that global style and local content informations play a vital role in detection tasks.

- We propose a novel adaptation framework for cross-domain object detection. A style-aware feature fusion method and two plug-and-play adaptation modules are designed to reposition the focus of the model to domain-specific features by restructuring the style and content of feature.

- We conduct extensive experiments in adaptation between both similar and dissimilar domains to validate the effectiveness of our approach. The compelling experimental results and visualizations demonstrate that our method significantly boost the performance of existing Domain Adaptive Faster R-CNN detectors under various domain-shift scenarios.

## 2 RELATED WORK

### 2.1 DOMAIN ADAPTATION

Domain adaptation (Tzeng et al., 2014; Gong et al., 2015; Sun & Saenko, 2016; Zou et al., 2018) with convolution neural network has emerged as a popular paradigm to boost the performance of trained model when training and test data do not satisfy the assumption of independent identical

distribution. They strive to learn domain-invariant features, and along this line the methods can be roughly divided into two categories: statistical matching and adversarial learning. The common idea of statistical matching (Fernando et al., 2014; Gong et al., 2015; Zellinger et al., 2017; Peng et al., 2018) is to measure and minimize the distance between latent feature space of different domains by appropriate metrics. The methods (Chang et al., 2019; Chen et al., 2018a; Russo et al., 2017; Fu et al., 2020; Xu et al., 2020) based on adversarial learning aim to learning domain-invariant feature by construct minimax optimization with the domain classifier. In addition, Li et al. (2018) proposed to modulate the statistics in all BN layers across the network. And He & Zhang (2019) applied a progressive curriculum-based adversarial training to boost fine-grained visual categorization. Besides, Jin et al. (2021) designed a SNR module to ensure both generalization and discrimination capability. Despite the great success of the methods mentioned above, most of them are hard to be applied to object detection directly because of the local nature of detection tasks.

## 2.2 DA FOR OBJECT DETECTION

Recently, an increasing number of researches (Chen et al., 2018b; Zhu et al., 2019; Yang et al., 2020; Zhao et al., 2020) have been proposed for cross-domain object detection. Domain Adaptive Faster R-CNN (Chen et al., 2018b) proposed to learn domain-invariant features in image-level and instance-level by adversarial learning. Furthermore, Saito et al. (2019) applied strong-weak hierarchical feature alignment to alleviate non-transferable features, and Zhu et al. (2019) repositioned the focus of adaptation from global to local and introduces an additional module to reconstruct the patches. For the sake of more accurate feature alignment, Xu et al. (2020) integrated an image-level multi-label classifier upon the detection backbone to pay more attention on the foreground instances and re-weight the proposals to automatically hunt for the hard samples. Similarly, Zheng et al. (2020) adopted the attention mechanism to highlight the importance of the foreground regions, and then align instances with the same category between source and target domain by maintaining global prototypes. Besides, Chen et al. (2020) hierarchically calibrated the transferability of feature representations in images/instance/local-region level, aiming to harmonize transferability and discriminability. And Vibashan et al. (2021) employed category-wise discriminators to ensure category-aware feature alignment. The above methods enable the network to extract domain-invariant features in virtue of refined feature alignment, and enhance the transferablility of the model. However, how to make better use of target-like domain-specific details is under-explored.

## 2.3 STYLE TRANSFER

Style transfer (Ulyanov et al., 2016; Yoo et al., 2019; Huang & Belongie, 2017; Johnson et al., 2016) has been proven efficient on the transformation of the object appearance. In general, style transfer is considered as a texture conversion issue. Some previous methods (Denton et al., 2015; Frigo et al., 2016) typically rely on low-level statistics but fail to extract the semantic image content. Gatys et al. (2016) pioneered the separation of image content information from its style and introduced a new algorithm to perform image style transfer. Furthermore, Ulyanov et al. (2017) propose a light-weight feed-forward network. Surprisingly, Ulyanov et al. (2016) achieves remarkable improvement by replacing BN with IN. Along this line, Dumoulin et al. (2016); Huang & Belongie (2017) go further and design alternative modules, which improve the quality and diversity of image stylization. And Kotovenko et al. (2021) review the stylization process and concentrate on optimizing parameterized brushstrokes. All these methods motivate us to design a complementary module for mining target-like details in domain adaptation. While previous works (Chen et al., 2020; Rodriguez & Mikolajczyk, 2019) employed CycleGAN to generate interpolation samples, they faced a critical limitation that the performance is greatly determined by image quality. In this paper, we propose to manipulate the content and style information simultaneously by feature reconstruction during the training phase, aiming to exploit more domain-specific features in an end-to-end fashion.

## 3 METHOD

### 3.1 PROBLEM FORMULATIONS

Unsupervised domain adaptation for object detection can boil down to the analysis of data distributions in two domains: a source domain with adequate annotated image data and a target domain

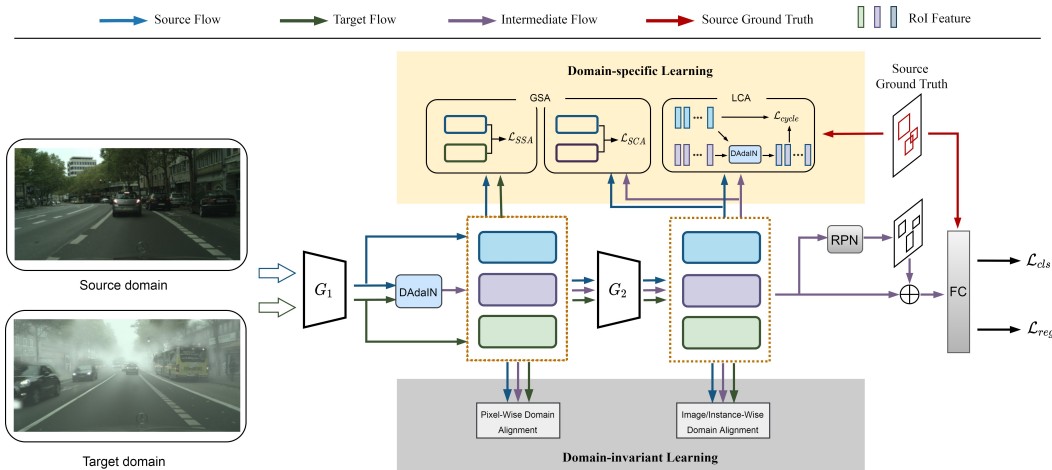

Figure 2: The Overview of our proposed adaptation framework. The main components of our model are DAdaIN, GSA and LCA. DAdaIN is inserted into the backbone network, which provides more domain-specific details through feature reconstruction. GSA consists of two sub-modules, i.e., Style Similarity Regularize and Style Consistency Regularize. They constrain the similarity and consistency of global image style to improve the robustness of the model. LCA highlights the importance of local instance content, and achieves the alignment without destroying domain-specific details.

with unlabeled images. Formally, we denote source domain as $\mathcal{D}_\mathcal{S} = \{(x_i^s, y_i^s)\}_{i=1}^{N_s}$, where $x_i^s$ and $y_i^s = (b_i^s, c_i^s)$ represent the $i$-th sample and label (class and bounding box) respectively. Similarly, target domain is defined as $\mathcal{D}_\mathcal{T} = \{(x_i^t)\}_{i=1}^{N_t}$, where $N_t$ is the amount of target samples. $\mathcal{D}_\mathcal{S}$ and $\mathcal{D}_\mathcal{T}$ have the identical categories tags $c$ but are drawn from diverse scenarios with heterogeneous data distribution. Our destination is to generalize the detector from labeled $\mathcal{D}_\mathcal{S}$ to unlabeled $\mathcal{D}_\mathcal{T}$.

### 3.2 FRAMEWORK OVERVIEW

The overview of proposed framework is illustrated in Figure 2. It consists of three main components: DAdaIN, GSA and LCA. Previous works (Hsu et al., 2020; Chen et al., 2020) exploited synthetic data by CycleGAN to narrow the domain gap, which cannot be trained in an end-to-end fashion and is sub-optimal for tackling the domain shift. In this paper, we design a style-aware feature fusion module called DAdaIN and insert it into the detector during the training phase aiming to improve the domain invariance of the model itself. Moreover, Global Style Alignment (GSA) and Local Content Alignment (LCA) are introduced to regularize the style and content information between multiple domains respectively. We suppose that image-level style and instance-level content play a vital role in detection tasks. Extensive experiments verify the effectiveness of our method.

Our framework effectively increases the learning ability of the model for domain-specific features. Meanwhile, the proposed module can be flexibly used as a supplement to existing SOTA methods and improve the generalization of the model without any modification to the inference process.

### 3.3 STYLE-AWARE FEATURE FUSION

With the increase of domain gap, existing UDA methods usually exhibit instability or adaptability degradation. It can be intuitively understood as the result of domain-specific features missing. We suppose that low-level variations, e.g. color, texture and edges, constitute a unique style which reflects the difference of feature distributions between different domains. Inspired by Huang & Belongie (2017), we adopt the style-aware feature fusion method and design Domain Adaptive Instance Normalization (DAdaIN) to provide extra target-like details. Our key insight is that while it is difficult to extract discriminative features in target domain, it is feasible to directly assign the underlying details to the model via feature style transfer, so as to alleviate the inconsistency between training and testing.

Specifically, we denote $f_i \in \mathbb{R}^{N \times C \times H \times W}$ as the feature map extracted by the detector in domain $i \in \{\mathcal{D_S}, \mathcal{D_T}\}$, where $H \times W$ and $C$ are the spatial dimensions and the number of feature map channels, respectively. DAdaIN adaptively computes the affine parameters from the target feature $f_t$ to scale and shift the source feature $f_s$:

$$
\begin{aligned}
\text{DAdaIN}(f_s, f_t) &= \sigma_t \left( \frac{f_s - \mu_s}{\sigma_s} \right) + \mu_t \\
&= \frac{\sigma_t}{\sigma_s} f_s + (1 - \frac{\sigma_t}{\sigma_s}) \mu_s + (\mu_t - \mu_s)
\end{aligned}
\tag{1}
$$

where $\mu_i \in \mathbb{R}^{N \times C}$ and $\sigma_i \in \mathbb{R}^{N \times C}$ represent the mean and standard deviation of each channel for each sample $f_i$.

From Eq.1, we can derive that the output feature of DAdaIN depends on the feature statistics ($\mu$, $\sigma$) computed across spatial dimensions. In order to enrich the diversity of synthesis feature and maintain the stability during training phase, we introduce $\kappa_1$ to control the translation scale between source and target domains. Meanwhile, we use $\kappa_2$ to balance the translation bias caused by the scale of parameter $\kappa_1$. The DAdaIN is adopted in the feature space of intermediate domain $\mathcal{D_M}$. Eq.1 with translation parameters can be re-written as:

$$
\text{DAdaIN}(f_s, f_t; \kappa_1, \kappa_2) = \kappa_1 \frac{\sigma_t}{\sigma_s} f_s + (1 - \kappa_1 \frac{\sigma_t}{\sigma_s}) \mu_s + \kappa_2 (\mu_t - \mu_s)
\tag{2}
$$

where $\kappa_1$ and $\kappa_2$ obey the normal distribution $\mathcal{N}(1, 1)$.

Finally, the style-aware synthesis feature $f_m$ can be defined as following:

$$
f_m = \lambda \, \text{DAdaIN}(f_s, f_t) + (1 - \lambda) f_s
\tag{3}
$$

where $\lambda$ is a hyper-parameter to control the degree of feature style transfer. In all experiments, we set $\lambda$ to 1.

Technically, we integrate DAdaIN module after $\text{relu4\_2}$ of VGG16, then fuse the style and content informations to generate the synthesis features $f_m$. It is worth noting that, all subsequent work is based on $f_m$. Moreover, since the role of DAdaIN is to introduce domain-specific details during training phase, it is feasible to remove such module for inference without affecting the performance.

By doing so, we can generate the synthesis features which encode the discriminative information of target domain in an end-to-end fashion. Besides, since the content of synthesis feature is shared with the source domain, the label information of the source domain can be better used.

### 3.4 DISCRIMINATIVE FEATURE MINING

**Global Style Alignment.** While the DAdaIN enrich the diversity of synthesis features, it also challenges the model's invariance to the re-style features in shallow layers. In order to further regularize the image style information across multiple domains, we propose Global Style Alignment (GSA) to constrain the style consistency with feature statistics.

GSA performs multi-level style alignment across different domains. It is composed by two parts: style similarity alignment and style consistency alignment. Style similarity alignment is applied on the low-level features across the source and target domain, which aiming to improve training stability. Style consistency alignment is applied on the high-level features across the synthesis and target domain to constrain the consistency of image style. Considering Saito et al. (2019) proposes that strong-weak feature alignment works well between both the similar and dissimilar domains, we construct different criteria to align the global style information at corresponding feature level.

Specifically, we define style similarity alignment as follows to measure the style similarity of shallow features between $\mathcal{D_S}$ and $\mathcal{D_T}$:

$$
\mathcal{L}_{SSA} = 2 - \sum_{v \in \{\mu, \sigma\}} e^{-\left| v(f_s^1) - v(f_t^1) \right|}
\tag{4}
$$

where the $f_s^1$ and $f_t^1$ are output features of the $G_1$. With the constraints on shallow feature statistics, the domain gap is further reduced, so that the model can be stably trained even on dissimilar domains.

Besides, style consistency alignment adopt smooth L1 loss to align the mean and variance of high-level features between the $\mathcal{D}_{\mathcal{M}}$ and $\mathcal{D}_{\mathcal{T}}$:

$$\mathcal{L}_{SCA} = \sum_{v \in \{\mu, \sigma\}} \text{smooth}_{L_1} \left( v(f_m^2), v(f_t^2) \right) \tag{5}$$

where the $f_m^2$ and $f_t^2$ are output features of the $G_2$. In other words, it promotes the model to extract stable feature in deep layers for the re-style instances by aligning the features of the target and intermediate domain. And as a result, the model's invariance to shallow features is improved.

**Local Content Alignment.** Attention to local regions which containing foreground objects is crucial to the success of object detection. SCDA (Zhu et al., 2019) repositions the focus of the cross-domain adaptation process, from global to local, and shows impressive performance. However, we find that strong local feature alignment will cause instability during training. We suppose that it is due to a mismatching of instance information. Strong local alignment will induce the model to pay more attention to common components. When the styles of instances in different domains differ greatly, the network will predict unexpected results and the training instability will be aggravated. A better way to enhance the invariance of the model to the local instance in various styles is aligning only the content information of the instance. Moreover, we find that GSA tends to result in negative style transfer because of the poor-performing Convolutional Neural Networks, especially in the early stage. Motivated by the aforementioned findings, we design Local Content Alignment(LCA) to further regularize the content of instances without destroying domain-specific details.

Specifically, we train RPN in the intermediate domain $\mathcal{D}_{\mathcal{M}}$ and extract the instance feature of local regions in source and intermediate domain. Since $\mathcal{D}_{\mathcal{M}}$ is generated by decoupling and reconstructing the content and style of $\mathcal{D}_{\mathcal{S}}$ and $\mathcal{D}_{\mathcal{T}}$, the content representation ought to be consistent between $\mathcal{D}_{\mathcal{S}}$ and $\mathcal{D}_{\mathcal{M}}$. In order to achieve a more accurate alignment of instance content information, we firstly re-style extracted instance feature in intermediate domain from synthesis to source:

$$f_{m'} = \text{DAdaIN}(f_m, f_s) \tag{6}$$

Then, we use cycle consistency loss to regularize the content of instances with unified style:

$$\mathcal{L}_{cycle} = \sum_{r \in GT} \text{smooth}_{L_1} \left( f_{m'}(r), f_s(r) \right) \tag{7}$$

where $f(r)$ denotes the feature of foreground region $r$. Considering that the ground truth is shared between the intermediate and source domain, we use the ground truth $GT$ to extract the foreground regions in $\mathcal{D}_{\mathcal{S}}$ and $\mathcal{D}_{\mathcal{M}}$. By means of LCA, the consistency of instance content before and after feature style transfer is maintained, and the feature misalignment can be avoided.

## 3.5 TOTAL OBJECTIVE FUNCTION

We denote the objective of detection task as $\mathcal{L}_{det}$, which consists of $\mathcal{L}_{cls}$ and $\mathcal{L}_{reg}$. $\mathcal{L}_{cls}$ aims to measure the classification accuracy, while $\mathcal{L}_{reg}$ is used to evaluates the location error. The DA Faster R-CNN series apply domain classifier in image and instance level to align the global and local feature. We denote the objective of domain classifier as $\mathcal{L}_{ADV}$:

$$\mathcal{L}_{ADV} = \sum_l \mathbb{E}_{f \sim \mathcal{D}_{\mathcal{S}}} \log D_l(f) + \sum_l \mathbb{E}_{f \sim \{\mathcal{D}_{\mathcal{T}}, \mathcal{D}_{\mathcal{M}}\}} \log(1 - D_l(f)) \tag{8}$$

where domain discriminator $D_l$ (element-wise/image-wise/instance-wise) promotes domain-invariant feature extraction with respect to the base conv $G$ shown in Figure 2.

We consider our approach as a complementary module which can be integrated into the existing DA Faster R-CNN series. The overall objective is:

$$\mathcal{L}_{total} = \mathcal{L}_{det} + \mathcal{L}_{ADV} + \alpha \mathcal{L}_{GSA} + \beta \mathcal{L}_{LCA} \tag{9}$$

where $\alpha$ and $\beta$ are trade-off factors for GSA and LCA module.

## 4 EXPERIMENTS

### 4.1 IMPLEMENTATION DETAILS

For all the experiments, we adopt the Faster R-CNN with VGG16 (Simonyan & Zisserman, 2014) or ResNet-101 (He et al., 2016) backbone. The weights of the network is initialized with the pre-trained model on ImageNet. The hyper-parameters of the model is set following Chen et al. (2018b); Xu et al. (2020); Zheng et al. (2020); Chen et al. (2020), and the shorter side of all training and test datasets is resized to 600 pixels. We use the stochastic gradient descent SGD (Bottou, 2010) as the optimizer with an initial learning rate of $10^{-3}$ in the training phase, which is reduced to $10^{-4}$ after 50k iterations. We report the mAP with an IoU threshold of 0.5 to evaluate the model in test procedure. All these experiments are implemented by the PyTorch framework.

### 4.2 COMPARISON RESULTS

We conduct quantitative comparsion on three cross-domain detection tasks by means of five datasets. Due to the limit to getting some papers open source code, we only employ SWDA (Saito et al., 2019) and HTCN (Chen et al., 2020) as baseline models and straightly integrate our approach into them. For the sake of fairness, all of the SOTA models we cite are based on the Faster R-CNN with same backbone network. When comparing with them, all results we report are cited from original papers. Besides, we trained the model on the labeled source and target domain respectively, which are treated as the upper and lower bound of the algorithm.

**Normal to Foggy.** Changes in weather conditions pose a major challenge to the robustness of algorithm in real world. We thus perform the adaptation from Cityscape (Cordts et al., 2016) to Foggy Cityscape (Sakaridis et al., 2018) aiming to verify the domain invariance of algorithm to weather changes. Cityscapes contains 2,975 training images and 500 test images with dense pixel-level annotations. It is collected from the outdoor street scenarios under normal weather condition in German cities, while Foggy Cityscapes derives from Cityscapes and uses depth information to simulate foggy scenario. The results of the adaptation are reported in Table 1. Our proposed method achieves state-of-the-art mAP and boost the performance of SWDA and HTCN with 3.6% and 2.6% respectively. The compelling results demonstrate that our method effectively improve the generalization of the SOTA methods to target domain by restructuring the style and content of feature representations.

Table 1: Results (%) of different methods on adaptation from Cityscapes to Foggy-Cityscapes. "Source Only" denotes the Faster R-CNN model trained on the source domain only. "Oracle" represents the model trained on the labeled target domain. VGG16 is used as the backbone network.

| Method | bus | bicycle | car | mcycle | person | rider | train | truck | mAP |
|---|---|---|---|---|---|---|---|---|---|
| Source Only | 20.8 | 31.3 | 33.5 | 18.1 | 24.9 | 32.8 | 9.1 | 10.4 | 22.6 |
| SCDA (Zhu et al., 2019) | 39.0 | 33.6 | 48.5 | 28.0 | 33.5 | 38.0 | 23.3 | 26.5 | 33.8 |
| CTF (Zheng et al., 2020) | 43.2 | 37.4 | 52.1 | 34.7 | 34.0 | 46.9 | 29.9 | 30.8 | 38.6 |
| DAA (Fu et al., 2020) | 46.6 | 36.9 | 48.8 | 34.0 | 33.2 | 47.6 | 38.2 | 28.1 | 39.2 |
| MeGA (Vibashan et al., 2021) | **49.2** | 39.0 | 52.4 | 34.5 | **37.7** | **49.0** | 46.9 | 25.4 | 41.8 |
| SWDA (Saito et al., 2019) | 36.2 | 35.3 | 43.5 | 30.0 | 29.9 | 42.3 | 32.6 | 24.5 | 34.3 |
| SWDA+ours | 42.8 | 37.5 | 49.5 | 33.3 | 34.0 | 45.9 | 33.0 | 27.5 | 37.9 |
| HTCN (Chen et al., 2020) | 47.4 | 37.1 | 47.9 | 32.3 | 33.2 | 47.5 | **40.9** | 31.6 | 39.8 |
| HTCN+ours | 49.1 | **40.2** | **52.6** | **37.2** | 37.3 | 48.8 | 40.8 | **33.0** | **42.4** |
| Oracle | 53.7 | 41.4 | 53.3 | 39.1 | 38.6 | 49.5 | 38.7 | 35.8 | 43.8 |

**Dissimilar Domains Adaptation.** Adaptation between dissimilar domains has been a difficult point in recent years. Models are always in trouble with the adaptation as the domain shift increases. Followed Saito et al. (2019), we utilize Pascal VOC (Everingham et al., 2010) as the source domain and either the Clipart1k (Inoue et al., 2018) or Watercolor (Inoue et al., 2018) as target domain to further experiment on dissimilar domains. PASCAL VOC contains 20 common categories of images and their bounding box. Clipart1K is a cartoon image dataset which contains 1K images in total. It

has the same category as PASCAL VOC, but there is a large domain shift between the same category. Watercolor is similar to Clipart1K, which contains 6 categories in common with PASCAL VOC and 2K painting images in total. ResNet101 (He et al., 2016) pre-trained on ImageNet (Krizhevsky et al., 2012) is adopted as our backbone network.

**1) Results on Clipart.** Table 2 shows the results of adaptation from PASCAL VOC to Clipart1K. It can be observed that our method improves SWDA and HTCN by 4.5% and 3.0% respectively. The superiority of the proposed method demonstrate the effectiveness of feature fusion even on dissimilar domains. In addition, we noticed that the performance of some classes is lower than baseline. We suppose it is mainly due to the difference of inter-class feature distribution. As the changes of object shape, size, texture, etc., its challenging to use the image-level feature statistics to represent the discriminative information of all categories of objects accurately. But for most categories, our method is effective and can bring performance improvements.

Table 2: Results (%) on adaptation from PASCAL VOC → Clipart Dataset. ResNet101 is used as the backbone network.

| Method | aero | bcycle | bird | boat | bottle | bus | car | cat | chair | cow | table | dog | hrs | bike | prsn | plnt | sheep | sofa | train | tv | mAP |
|---|---|---|---|---|---|---|---|---|---|---|---|---|---|---|---|---|---|---|---|---|---|
| Source Only | 35.6 | 52.5 | 24.3 | 23.0 | 20.0 | 43.9 | 32.8 | 10.7 | 30.6 | 11.7 | 13.8 | 6.0 | 36.8 | 45.9 | 48.7 | 41.9 | 16.5 | 7.3 | 22.9 | 32.0 | 27.8 |
| SCL (Dong et al., 2020) | 33.4 | 49.2 | **36.0** | 27.1 | 38.4 | 55.7 | 38.7 | 15.9 | 39.0 | 59.2 | 18.8 | 23.7 | 36.9 | 70.0 | 60.6 | 49.7 | 25.8 | **34.8** | 47.2 | 51.2 | 40.6 |
| CRDA (Xu et al., 2020) | 28.7 | 55.3 | 31.8 | 26.0 | 40.1 | 63.6 | 36.6 | 9.4 | 38.7 | 49.3 | 17.6 | 14.1 | 33.3 | 74.3 | 61.3 | 46.3 | 22.3 | 24.3 | 49.1 | 44.3 | 38.3 |
| DAA (Fu et al., 2020) | 35.0 | 59.5 | 34.6 | 30.2 | 38.1 | 60.2 | 40.2 | **20.5** | 39.3 | 58.5 | **26.4** | 22.8 | 33.8 | 82.9 | **64.4** | 48.8 | 18.0 | 28.6 | **57.6** | 46.2 | 42.3 |
| SWDA (Saito et al., 2019) | 26.2 | 48.5 | 32.6 | 33.7 | 38.5 | 54.3 | 37.1 | 18.6 | 34.8 | 58.3 | 17.0 | 12.5 | 33.8 | 65.5 | 61.6 | **52.0** | 9.3 | 24.9 | 54.1 | 49.1 | 38.1 |
| SWDA+ours | 40.1 | **62.6** | 27.0 | 39.1 | 42.9 | 65.7 | 39.6 | 17.1 | 36.3 | 62.1 | 18.0 | 24.2 | 44.7 | **85.9** | 64.2 | 41.7 | 33.3 | 22.8 | 35.9 | 48.9 | 42.6 |
| HTCN (Chen et al., 2020) | 33.6 | 58.9 | 34.0 | 23.4 | **45.6** | 57.0 | 39.8 | 12.0 | **39.7** | 51.3 | 21.1 | 20.1 | 39.1 | 72.8 | 63.0 | 43.1 | 19.3 | 30.1 | 50.2 | **51.8** | 40.3 |
| HTCN+ours | **43.6** | 58.8 | 32.1 | **39.6** | 39.8 | **68.5** | **45.3** | 11.4 | 38.1 | **62.4** | 20.9 | **30.3** | **46.7** | 76.5 | 62.9 | 39.3 | **33.6** | 23.5 | 48.1 | 45.3 | **43.3** |

**2) Results on Watercolor.** According to Table 3, we observe that proposed method reaches the mAP of 57.0% and 55.2% with a gain of 3.7% and 2.0% over the baseline on the adaptation from PASCAL VOC to Watercolor. The fact indicates that our method help the baseline model capture the style information of Watercolor and converge to better performance.

Table 3: Results (%) on adaptation from PASCAL VOC to Watercolor Dataset. ResNet101 is used as the backbone network.

| Method | bcycle | bird | car | cat | dog | prsn | mAP |
|---|---|---|---|---|---|---|---|
| Source Only | 68.8 | 46.8 | 37.2 | 32.7 | 21.3 | 60.7 | 44.6 |
| DA-Faster (Chen et al., 2018b) | 75.2 | 40.6 | 48.0 | 31.5 | 20.6 | 60.0 | 46.0 |
| DAA (Fu et al., 2020) | 85.1 | **56.6** | 46.2 | 39.9 | 36.9 | 65.6 | 55.1 |
| CDG (Li et al., 2021a) | **97.7** | 53.1 | 52.1 | 47.3 | 38.7 | **68.9** | **59.7** |
| SWDA (Saito et al., 2019) | 82.3 | 55.9 | 46.5 | 32.7 | 35.5 | 66.7 | 53.3 |
| SWDA+ours | 84.8 | 51.6 | 51.4 | **47.9** | **38.9** | 67.3 | 57.0 |
| HTCN (Chen et al., 2020) | 77.4 | 52.2 | **56.1** | 36.3 | 33.5 | 63.4 | 53.2 |
| HTCN+ours | 76.2 | 52.2 | 52.2 | 45.9 | 37.5 | 66.9 | 55.2 |

## 4.3 FURTHER EMPIRICAL ANALYSIS

**Ablation Study.** In this session, we conducted ablation experiments to validate the effectiveness of proposed method. Firstly, we explore the role of each component we proposed. Specifically, we append each module gradually and observe the performance of the model on Normal→Foggy. The results are shown in Table 4. It can be found that the performance increases when any component is appended, which proves that all component are designed reasonably and contribute to the adaptation. Secondly, in order to verify that discriminative feature mining is helpful to improve the stability and invariance of the model, we adjust the weight $\lambda$ in Eq.3 to control the degree of feature style transfer. We change $\lambda$ from 0 to 1 gradually and observe the performance of the model on Normal→Foggy. As shown in Figure 3 (a), we find that the mAP continuously increases with the style weight $\lambda$, which proves the validity of DAdaIN and highlights the role of discriminative feature in domain adaptation. Thirdly, we evaluate the effects of the parameter $\alpha$ and $\beta$ in Eq.9 which balances the

Table 4: Ablation of proposed method on Cityscapes → Foggy Cityscapes. *DIN* represents the proposed DAdaIN, and *SS*, *SC*, *LC* indicate style similarity alignment, style consistency alignment, and local content alignment.

| Method | DIN | SS | SC | LC | mAP | Gain |
|---|---|---|---|---|---|---|
| HTCN | | | | | 39.8 | - |
| Proposed | ✓ | ✓ | | | 40.3 | 0.5 |
| | ✓ | | ✓ | | 40.4 | 0.6 |
| | ✓ | | | ✓ | 40.8 | 1.0 |
| | ✓ | ✓ | ✓ | | 41.2 | 1.4 |
| | ✓ | ✓ | | ✓ | 41.5 | 1.7 |
| | ✓ | | ✓ | ✓ | 41.7 | 1.9 |
| | ✓ | ✓ | ✓ | ✓ | 42.4 | 2.6 |

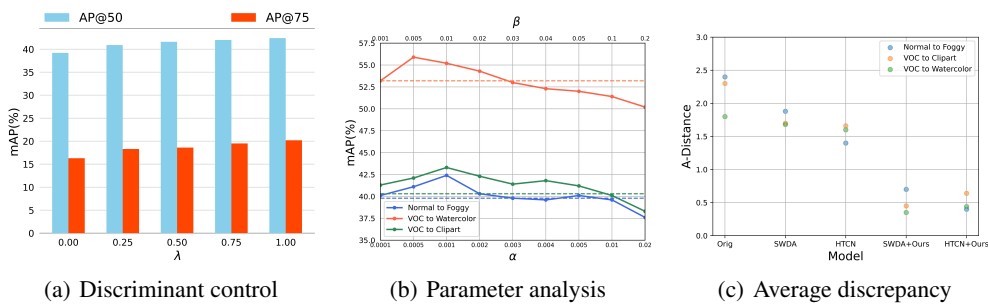

(a) Discriminant control     (b) Parameter analysis     (c) Average discrepancy

Figure 3: Analysis of discriminative feature and hyper-parameter on multi task.

contribution of GSA and LCA, respectively. As shown in Figure 3 (b), we change the parameters and compare our method with the baseline HTCN on different scenarios, which are drawn with solid and dashed lines. We can find the performance first rises and then drops with the increase of $\alpha$ and $\beta$, which proves that proper trade-off can help boost the performance.

**Domain Discrepancy.** To measure the cross-domain representation discrepancy, we introduce the proxy $\mathcal{A}$-distance (Schlkopf et al.) which is defined as $d_{\mathcal{A}} = 2(1 - 2\epsilon)$. The variable $\epsilon$ denotes the generalization error on the binary classification problem of discriminating the source and target domain. As shown in Figure 3 (c), compared to baseline model, our method further narrows the discrepancy between source and target domain, which demonstrate that discriminative features introduce key information that cannot be brought by domain invariant features. Furthermore, we visualize the image features learned for the adaptation on multi task using t-SNE (Van der Maaten & Hinton, 2008). Due to the limit of space, the results are shown in appendix.

**Detection Examples.** Figure 4 displays the qualitative detection results on adaptation tasks. Our method outperform the Source Only and HTCN baseline method. Even for the corner cases, e.g. occlusion and small object, our method still makes accurate predictions and inhibits false positives, which proves that our approach successfully help the model focus on domain-specific details.

## 5 CONCLUSION

In this work, we propose a simple yet effective framework for cross-domain object detection. The proposed framework consists of three key components: DAdaIN, GSA and LCA. DAdaIN supplement the domain-specific representation by restructuring the content and style information. GSA and LCA are exploited to regularize the consistency of style and content between multi domains. Extensive experiments show that our method can be used as a complementary module to highlight target-like details and improve the performance of existing methods on several benchmark datasets.

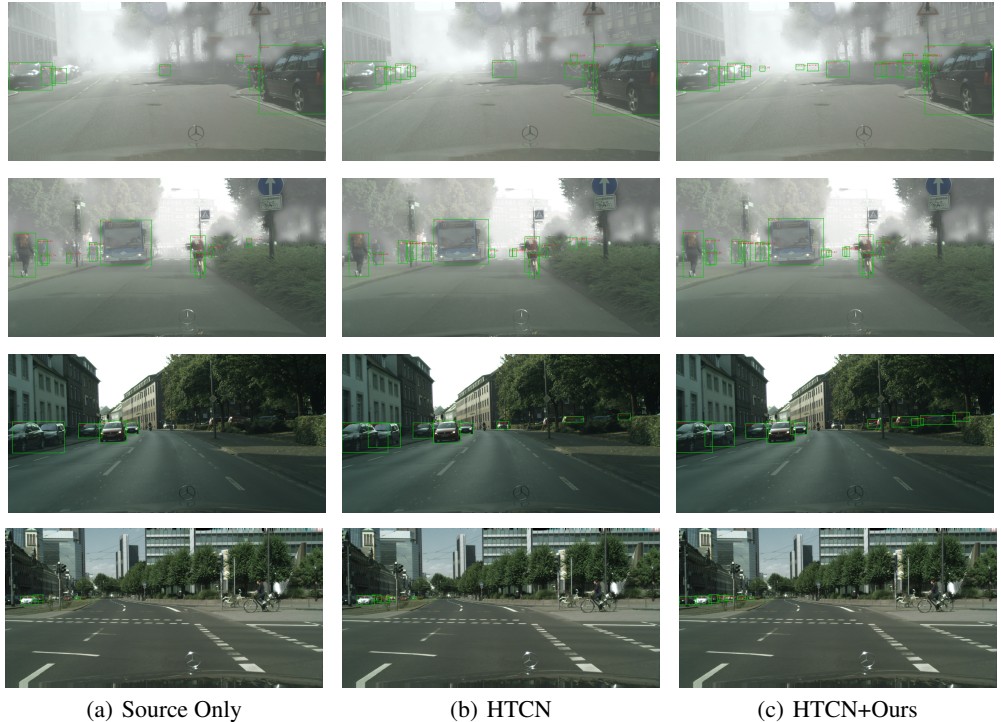

|                   |                |                   |
| :---------------: | :------------: | :---------------: |
| (a) Source Only   | (b) HTCN       | (c) HTCN+Ours     |

Figure 4: Qualitative detection results on the target domain. First and second rows: Normal→Foggy adaptation scenario. Third and fourth rows: Sim10K→Cityscapes adaptation scenario.

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

# A APPENDIX

## A.1 MORE ABLATION STUDIES

In this session, we present the additional results with different settings of GSA and LCA in Table 5 and 6 on the adaptation from Cityscape to Foggy Cityscape.

### A.1.1 ANALYSIS OF GSA

As shown in Table 5, GSA-A applies smooth L1 loss on shallow features to regularize the style similarity. And GSA-B represents that the L2 loss is applied on high-level features. We can find that the above methods achieve sub-optimal performance in the adaptation task.

Table 5: Results (%) on adaptation from Cityscape to Foggy Cityscape Dataset.

| Method | bus | bike | car | mcycle | prsn | rider | train | truck | mAP |
|---|---|---|---|---|---|---|---|---|---|
| Baseline | 48.8 | 35.6 | 51.6 | 31.1 | 36.1 | 47.8 | 41.5 | 30.6 | 40.4 |
| GSA-A | 45.8 | 39.5 | 52.7 | 36.3 | 38.3 | 47.9 | 34.5 | 27.2 | 40.3 |
| GSA-B | 45.8 | 40.2 | 52.6 | 36.9 | 38.6 | 48.5 | 36.9 | 31.7 | 41.4 |
| GSA | 49.1 | 40.2 | 52.6 | 37.2 | 37.3 | 48.8 | 40.8 | 33.0 | 42.4 |

### A.1.2 ANALYSIS OF LCA

Table 6 shows the results for three different variants of the LCA. LCA-A extracts the features in proposals generated by RPN rather than ground truth. LCA-B regularizes the content information after RoI alignment module. And LCA-C achieves content alignment based on the de-normalized features of instances between source and target domains.

It can be observed that the content alignment based on proposals will cause some performance degradation. One possibility is that poor-performing networks causes RPN to produce more low-quality bounding boxes, which weakens the model's attention to the foreground instance. Moreover, the content alignment based on the de-normalized features can produce similar results to LCA, which prove the validity of local content alignment again.

Table 6: Results (%) on adaptation from Cityscape to Foggy Cityscape Dataset.

| Method | bus | bike | car | mcycle | prsn | rider | train | truck | mAP |
|---|---|---|---|---|---|---|---|---|---|
| Baseline | 46.1 | 40.1 | 52.6 | 33.1 | 37.3 | 48.1 | 41.7 | 28.8 | 41.0 |
| LCA-A | 43.2 | 38.4 | 51.9 | 33.1 | 36.2 | 46.8 | 31.9 | 30.5 | 39.0 |
| LCA-B | 49.9 | 39.1 | 52.6 | 32.8 | 37.5 | 47.0 | 30.5 | 31.9 | 40.2 |
| LCA-C | 49.3 | 40.7 | 52.8 | 39.9 | 38.3 | 49.1 | 36.0 | 32.1 | 42.3 |
| LCA | 49.1 | 40.2 | 52.6 | 37.2 | 37.3 | 48.8 | 40.8 | 33.0 | 42.4 |

## A.2 MORE FEATURE VISUALIZATIONS

In order to further analyze the significance of characteristic statistics and the role of DAdaIN, we visualized the features and statistics in different adaptation scenarios. We applied global average pooling on the low-level features extracted by different models. As shown in 5, our method effectively reduce the domain gap from the Oracle, which proves our method can help the model to extract more domain-specific features and improves the invariance of the model for domain shift. From Figure 6, we can observe that: 1) The distribution of the features is similar to their statistical values, which proves that the statistics have certain statistical significance. 2) After applying DAdaIN, the difference of feature distribution between the intermediate and target domain is further optimized, which demonstrate the validity of DAdaIN.

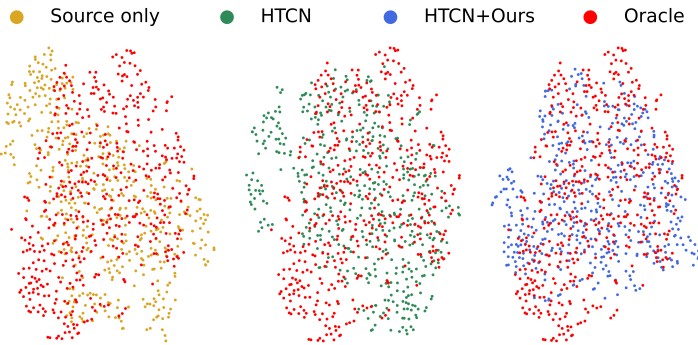

Figure 5: The t-SNE visualization of image features obtained from different models. Colors represent different models. We can see that the difference of feature distribution is further optimized.

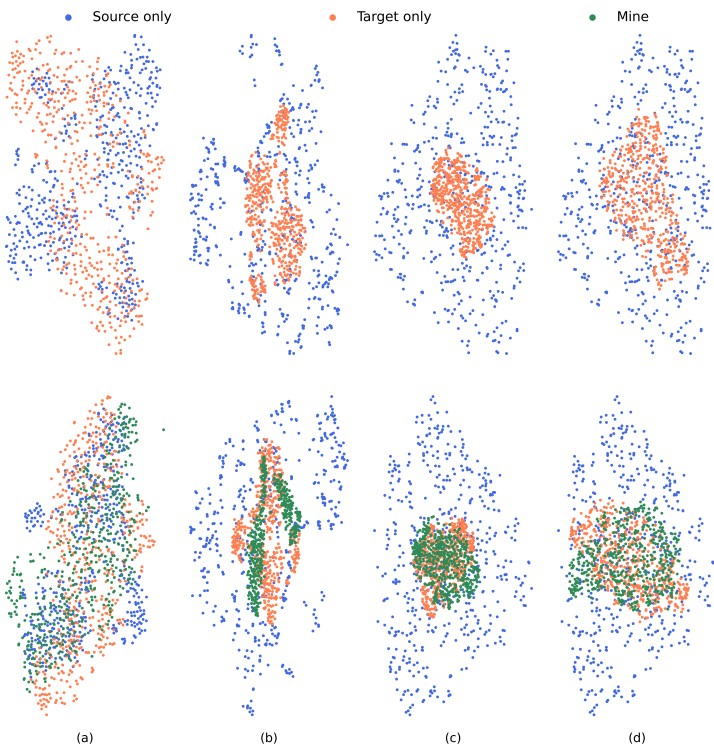

Figure 6: Visualizations of shallow features and statistics obtained from different models. "Source only" denotes the baseline model trained and tested only in source domain without adaptation. "Target only" represents the model trained and tested in target domain without adaptation. And "Mine" represents the model trained with our proposed method. It should be noticed that, when testing our method, we retain the DAdaIN module and extract the synthesis feature in intermediate domain to compare the feature distributions with source and target domain. First Row: The statistics of shallow features. Second Row: The shallow features obtained by applying global average pooling to the shallow features. Each column (from left to right) represents the model trained on Cityscape to Foggy Cityscapes, SIM10K to Cityscape, PASCAL VOC to Clipart, PASCAL VOC to Watercolor.

### A.3 MORE DETECTION RESULTS

We show more detection results on PASCAL VOC→Clipart and PASCAL VOC→Watercolor. As shown in Figure 7, our method promotes more accurate results than others, which prove the generalization of our method under different tasks.

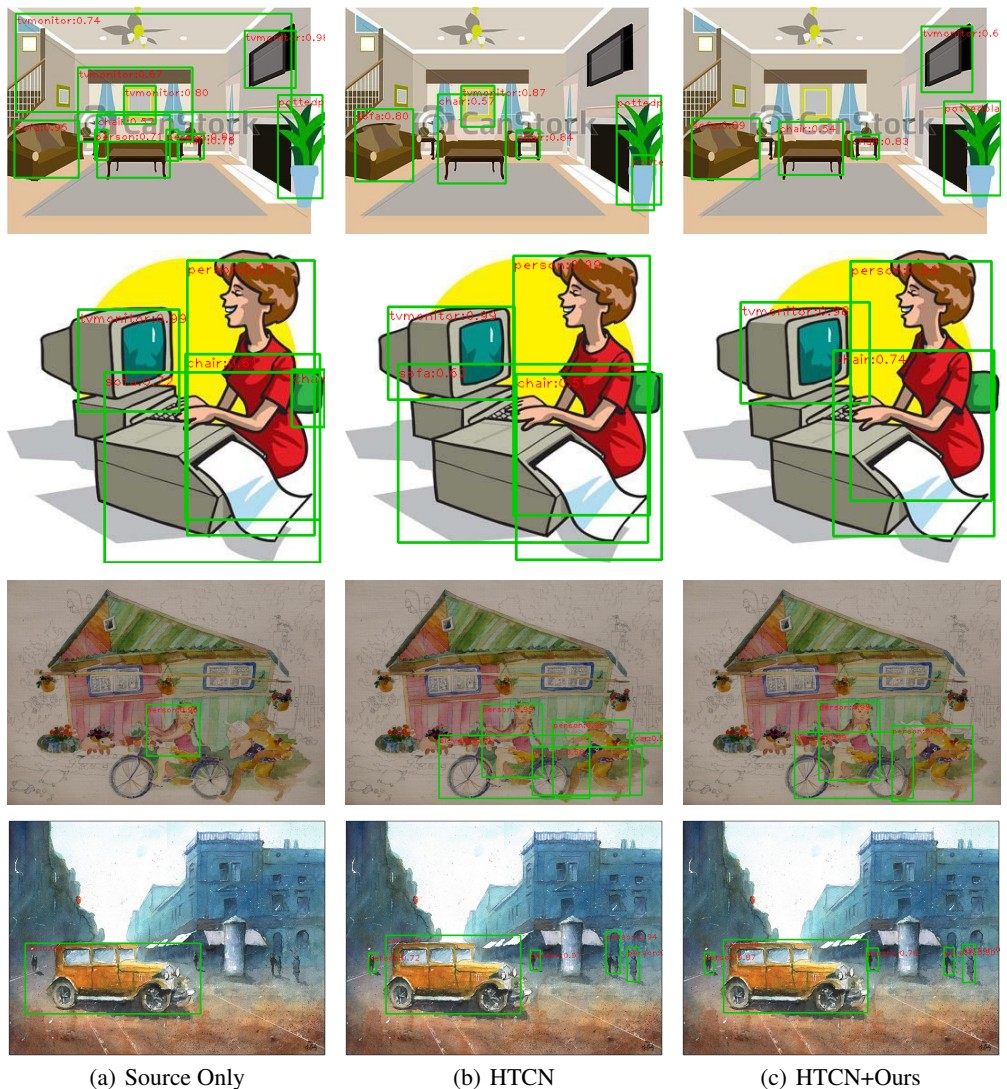

(a) Source Only       (b) HTCN       (c) HTCN+Ours

Figure 7: Qualitative detection results on the target domain. First and second rows: PASCAL VOC→Clipart adaptation scenario. Third and fourth rows: PASCAL VOC→Watercolor adaptation scenario.

