# OpenReview forum: "Decouple and Reconstruct: Mining Discriminative Features for Cross-domain Object Detection"
_ICLR.cc/2022/Conference — ICLR 2022 Submitted_

### Official Review · Reviewer_1yU8 · 2021-10-27

**Correctness:** 4
**Technical Novelty And Significance:** 3
**Empirical Novelty And Significance:** 4
**Recommendation:** 5
**Confidence:** 3

**Main Review:**

Strengths:
 + This paper introduces a new direction for cross-domain object detection. This is an important issue given the fact that “domain shift” occurs widely in the real-life applications and annotations are very costly to get. The proposed method borrows idea from style transfer and tries to reconstruct the content and style information without the requirement of new annotations. I feel this idea is interesting.

 + The proposed method is technically sound. Global style alignment and local content alignment have been carefully considered. I don't find any significant technical flaws in the technical part.

 + The proposed method achieves promising results across multiple benchmarks.

Weaknesses:

-The weakness part is that the proposed method generally adopts existing techniques and combines them to deal with the cross-domain object detection. It is still hard for me to find some novel technical contributions. I like the key idea, motivations and writing, yet I feel its technical contributions are still incremental. For example, the style feature fusion is from Huang&Belongie(2017), the global style alignment seems borrowed from Saito et al(2019) and local content alignment shares the similar idea with Zhu et al(2019). It would be great if the authors can further highlight/enhance the novel part to make the paper more solid.

-While the idea of learning domain-specific representation information is interesting and intuitively sensible, it seems speculative to me. It is difficult for readers to know what information has been captured. Some visualization or a more carefully targeted experiment showing what are in the domain-specific features are needed in order to justify the explanation that the authors give for their method.

-The overall performance is acceptable yet not impressive. The proposed approach achieves only slightly better performance than other approaches as shown in Table 2, or lower accuracy than SOTA method (Table 3). Can the authors further provide the FLOPs, memory costs and parameters of all compared methods to better show the superiority of the proposed approach compared to other alternatives.

**Summary Of The Paper:**

This paper presents a new method for cross-domain object detection, which employs the style-aware feature fusion method and two novel modules to build the model. The key idea of this paper is to assign the underlying details to the model by using techniques from style transfer to overcome the domain gap. The proposed method is evaluated on multiple datasets and achieves promising results.

**Summary Of The Review:**

This paper introduces a new approach for cross-domain object detection but is somewhat weak in technical contributions.

---

### Official Review · Reviewer_VfW6 · 2021-10-31

**Correctness:** 2
**Technical Novelty And Significance:** 2
**Empirical Novelty And Significance:** 2
**Recommendation:** 5
**Confidence:** 3

**Main Review:**

Strengths:
(1) The paper is well written and easy to understand.
(2) The extensive experiments show the effectiveness of their proposed modules.

Weakness:
(1) The unclear motivation. The paper points out that image-level style and instance-level content matter in cross-domain detection. However, I do not understand the explanation that "The style information should be globally consistent within its own domain while local homogeneous content information between different domains ought to be analogous, and vice versa". In my opinion, the better extraction for the domain-specific features could lead to better decoupling the domain-invariance and domain-specific features. Thus, the domain-invariance feature could be enhanced. It is just my personal viewpoint and might be wrong. Please the author clarifies my questions.

(2) Some missing illustrations for the framework. For example, what's G_1 and G_2? Where is the L_adv in the Figure2?

(3) Is it wrong in the last row of Figure4? the nearest cyclist is not detected.

**Summary Of The Paper:**

This paper proposes a novel adaptation framework to balance transferability and discriminability for cross-domain object detection. Concretely, they focus on domain-specific features thus manage to reconstruct the style and content of features by AdaIN and some alignment techniques. Extensive experiments show the performance improvement by proposed modules.



**Summary Of The Review:**

Overall, I am wondering about the entire motivation of this paper. I acknowledge the strength of the proposed modules. However, these modules lack some novelties in my opinion. They are like the combination of some typical modules in a generative model. Thus I tend to choose marginally below the acceptance threshold.

I would like to upgrade the rating if the author addressed my concerns.

---

> ### Author Response · Authors · 2021-11-18
> **Clarifications**
>
> **QA1: The Unclear Motivation .**
> In recent years, most SOTA methods focus on the domain-invariant feats by adversarial training. They made great progress but pay little attention to domain-specific feats. We aim to  introduces a new direction for cross-domain object detection to prompt further development.
> In most cases, there are differences in data distribution between different scenarios. Meanwhile, the distribution of data collected in the same scene is more uniform. Only mining for the common attributions is not enough for UDA because the domain-specific information in target can lead to a better performance when we aim to develop the performance in target domain. We suppose that feature style is strongly related to the domain-specific information, and globally consistent within its own domain and different between source and target domain. So we use global feature statistics to describe the variety. As for instance content, it should be always uniform because whether in the United States or Germany, day or night, the content of car should not be changed. This must be handled carefully during feature alignment. If we directly reduce the difference between local features, which has been used in many methods, we can only achieve sub-optimal results because the car in different domain is under different styles. So we propose LCA to remove global style and then align the features' content only. This is our key insight and motivation. As for the method, we introduce intermediate domain and conduct cycle style transform game in multi level to realize our ideas. This make our method different from others. We hope the insights can bu shared by more researchers.
> **QA2:  Some missing illustrations .**
> That is indeed negligence. *G1* and *G2* are different parts of backbone. And *Ladv* represents *Domain-invariant Learning* in Fig.2.
> **QA3: The nearest cyclist is not detected .**
> For the sim2cs task, we follow other works and detect the car only. That is our missing, we will further explain later.

---

### Official Review · Reviewer_FLN9 · 2021-11-02

**Correctness:** 2
**Technical Novelty And Significance:** 3
**Empirical Novelty And Significance:** 2
**Recommendation:** 5
**Confidence:** 4

**Main Review:**

The paper tackles a quite interesting and challenging problem that has received limited attention by the community in the recent past. The idea of building upon the concepts of style transfer to capture and encode domain-specific and domain-invariant features is interesting and sounding. I also appreciated the fact that the made choices seems to be properly motivated. Despite such positive aspects, the paper seems to have some consistent presentation issues that make it hard to read and properly follow. There are unclear notations and cryptic sentences that act against a proper understanding of the proposed approach. There are also important claims that are not properly supported by empirical evidence.


Positive aspects:
+ The paper tackles a quite interesting and challenging problem that has received limited attention. The problem of being able to perform UDA for detecting objects belonging to the same classes under different domains has severe impacts on many applications ranging from security to autonomous driving, etc. The motivations behind the work seem to be solid and properly defined.
+ Building upon the style transfer ideas is an interesting approach. Leaving out more complex adversarial schemes to encode the domain-specific and domain-invariant features in source and target domain via an end-to-end trainable solution is quite appealing. The "simplicity" of the approach and the fact that it can be added to other existing methods are relevant points in favor of the proposed work.
+ To show the benefits of the proposed approach, 3 benchmark UDA setups have been considered. The experimental results section report on the comparison with very recent works on the 3 setups, also considering the same/similar models to provide a fair comparison.

Negative aspects:
- First and foremost, the paper contains typos and grammar problems that make reading difficult, thus acting against a proper understanding of the key ideas and techniques proposed by the paper. The text is often leaving out relevant technical concepts, thus failing to provide clear explanations. The adopted notation is confusing. For instance:
    - in section 3 the text refers to $G_1$ and $G_2$ whose definitions and descriptions are missing. There is evidence of these in Figure 2 but no clear explanation on what these are in practice. Same applies for the various $f(\cdot)$'s. It would be quite helpful if a more appropriate description is provided by following figures that would help to understand "what" goes "where" and "how" it is used.
   -  in section 3, when presenting the proposed DAdaIn layer, it seems that the function is depending on four inputs, with two of which represent the feature embedding from source and target, and the remaining two (i.e., $\kappa$'s) being hyperparameter controlling the translation scale and bias. It is not clear how these two last ones are obtained.
    - When presenting the achieved results, there are unclear subdivisions in the Tables. What do the double lines splitting the works used for comparison mean? No description is present in the caption nor in the text. Please clarify.
- There are claims in the paper that are not properly supported by experimental results or empirical evidence: while describing the local content alignment, the text states that "we find that strong local feature alignment will cause instability during training.". The text continues with: "GSA tends to result in negative style transfer because of the poor-performing Convolutional Neural Networks.". These are all strong, yet not properly supported, statements. Given the importance of these claims, it is highly recommended that appropriate experimental results are conducted (or results reported if already done).
- An obscure part of the paper regards the applicability of the proposed approach to existing methods. The text mainly refers to the faster R-CNN model with a VGG/ResNet backbone, then to the HTCN and SWDA methods, that, per-se are already doing domain adaptation for the object detection problem. A more detailed description of how the proposed method works alongside these ones is fundamental.


**Summary Of The Paper:**

The paper introduces a method to tackle the unsupervised domain adaptation problem, focusing on the object detection problem. Towards the final objective, an architecture building on the style transfer concepts is proposed. The main idea is to build upon the AdaIN ideas to encode the domain-specific and domain-invariant features in the source and target domain. Two additional modules are exploited to regularize the consistency of style and content between multiple domains. Experimental results on 3 benchmark UDA setups are conducted. Similar/better results than existing approaches are obtained.

**Summary Of The Review:**

In light of the aforementioned consideration, this reviewer believes that the paper has some merits but, the lack of a proper presentation, and the fact that there are not enough experimental results to support all paper claims, result in a submission that is not solid enough to justify an ICLR publication. These are two very important weaknesses that are likely to require more than a single major review round before the paper can have a proper shape. Thus, this reviewer is considering the submitted work below the acceptance bar but would be happy to change the opinion in light of more/proper evidence.

---

> ### Comment · Reviewer_FLN9 · 2021-11-25
> **Final thoughts and recommendations**
>
> Given the absence of any feedback from the authors regarding the raised doubts and questions, as well as, the feedback from fellow referees, I am keeping my initial recommendation, thus considering the current submission below the acceptance bar, despite the
> some merits that have to be acknowledged. The lack of a proper presentation, and the fact that there are not enough experimental results to support all paper claims, result in a submission that is not solid enough to justify an ICLR publication.

---

### Official Review · Reviewer_5g9D · 2021-11-07

**Correctness:** 3
**Technical Novelty And Significance:** 3
**Empirical Novelty And Significance:** 2
**Recommendation:** 5
**Confidence:** 2

**Main Review:**

+: The key assumption of the proposed method is reasonable, however, it lacks comparison to other methods under similar assumptions.

+: Experimental results show good performance of the proposed method.

+: It provides ablation study to show the effectiveness of different modules.

-: The core idea of the proposed method is to highlight the target-like discriminative feature, what is its difference to the self-training widely used in UDA, which after global aligning the source and target domains, selects pseo-samples for discriminative training in the target domain. Meanwhile, there is no discussion on related works of this aspect.

-: Experiments could be improved. The proposed method improves over SWDA and HTCN, however, they are not the SOTA. If the proposed method can be implemented on MeGA, it would be better. As for comparison of DA methods, the authors missed the self-training based DA methods, which also address the issue of local alignment/target discriminative.

-: The performance among different methods differs with respect to different adaptation settings, so, does the ablation study conducted on Normal -> Foggy is general enough for other adaptation scenarios? The difference can also be observed in Fig.3(b), where best parameter settings are different for different scenarios.

-: Some statement needs further clarification or validation.
(a) For the last sentence in the first paragraph of Section 1. it would also be helpful to show how many annotated images in the target domain are required (the same model trained on the source domain and finetuned in the target domain) to achieve the same performance as the best UDA methods. This will be a strong support to this argument. Otherwise, if the annotated images are few, the cost would not be high. Studying UDA for object detection would be less meaningful.
(b) In the end of page one, it mentions "it may implicitly drive networks to pay more attention on shared attributes and ignore the domain-specific feature", this could also be the problem of DA for other tasks. What is the uniqueness of DA for object detection? In other words, why should we focus on DA for object detection with the proposed method?
(c) the key assumption is that "image-level style and instance-level content" are important and can be decoupled. Is there any evidence to support this assumption. Another question is that  if it is still effective if we align instance-level and make the image-level keeping discriminative.

minor issue
the illustration in Fig.1 between DA and DA+Ours is hard to understand and less informative.

**Summary Of The Paper:**

This paper proposed a method for UDA object detection. The core idea is to increase the discriminative ability of learned features in the target domain while still globally aligning the feature distributions of the source and target domains. Under this motivation, the paper proposed modules of domain adapative instance normalization, global style alignment and local content alignment. Experiments are conducted on several benchmark adaptation scenarios, showing improvments on two existing methods.

**Summary Of The Review:**

The idea of the proposed method seems to be effective as supported by the experiments, however, it lacks disccusion and comparison to self-training methods that are based similar motivation, i.e., globally and locally alignment should be achieved simultaneously to achieve good DA performance. There are also many unclear issues that require further validation. Therefore, the paper is currently with more weaknesses compared to its strengths.

---

### Decision · Program_Chairs · 2022-01-20

**Decision:**

Reject

**Comment:**

This paper presents a method for unsupervised domain adaptation, focusing on the object detection problem. Under this framework, the paper proposes modules of domain adaptive instance normalization, global style alignment and local content alignment. The proposed method is evaluated on multiple datasets.

Several reviewers have pointed out that the paper lacks discussion and comparison to related methods. The paper has some merits but, the lack of a proper presentation, and the fact that there are not enough experimental results to support all claims in the paper, result in a submission that does not meet the bar of ICLR publication. Hence, the current paper is recommended to be not published at ICLR.